# Evaluation of Fecal Egg Count Tests for Effective Control of Equine Intestinal Strongyles

**DOI:** 10.3390/pathogens12111283

**Published:** 2023-10-26

**Authors:** Manigandan Lejeune, Sabine Mann, Holly White, Danielle Maguire, Jaime Hazard, Rebecca Young, Charles Stone, Doug Antczak, Dwight Bowman

**Affiliations:** 1Animal Health Diagnostic Center, Department of Population Medicine and Diagnostic Sciences, Cornell University, 240 Farrier Rd., Ithaca, NY 14853, USA; had24@cornell.edu (H.W.); ddm11@cornell.edu (D.M.); jlh57@cornell.edu (J.H.); rly23@cornell.edu (R.Y.); 2Department of Population Medicine and Diagnostic Sciences, College of Veterinary Medicine, Cornell University, Ithaca, NY 14853, USA; sm682@cornell.edu; 3College of Veterinary Medicine, Cornell University, Ithaca, NY 14853, USA; cbs257@cornell.edu; 4Department of Microbiology and Immunology, College of Veterinary Medicine, Cornell University, Ithaca, NY 14853, USA; dfa1@cornell.edu (D.A.); ddb3@cornell.edu (D.B.)

**Keywords:** fecal egg count, FEC, modified McMaster, Mini-FLOTAC, Wisconsin floatation, FEC gold standard

## Abstract

The American Association of Equine Practitioners strongly advocates evidence-based intestinal strongyle control in horses. It recommends targeted treatment of all heavy egg shedders (>500 eggs per gram (EPG) of feces), while the low shedders (0–200 EPG) are left untreated. As 50–75% of adult horses in a herd are low shedders, preventing them from unnecessary anthelmintic exposure is critical for tackling resistance. There are various fecal egg count (FEC) techniques with many modifications and variations in use, but none is identified as a gold standard. The hypothesis of the study was that the diagnostic performance of 12 commonly used quantitation methodologies (three techniques with four variants) differs. In this regard, method comparison studies were performed using polystyrene beads as proxy for intestinal strongyle eggs. Mini-FLOTAC-based variants had the lowest coefficient of variation (CV%) in bead recovery, whereas McMaster variants had the highest. All four variants of Mini-FLOTAC and the NaNO_3_ 1.33 specific gravity variant of modified Wisconsin followed a linear fit with R^2^ > 0.95. In contrast, the bead standard replicates for modified McMaster variants dispersed from the regression curve, causing a lower R^2^. The Mini-FLOTAC method seems less influenced by the choice of floatation solution and has better repeatability parameters and linearity for bead standard recovery. For FEC tests with high R^2^ (>0.95) but that underestimated the true bead count, a correction factor (CF) was determined to estimate the true count. Finally, the validity of CF was analyzed for 5 tests with R^2^ > 0.95 to accurately quantify intestinal strongyle eggs from 40 different horses. Overall, this study identified FEC methodologies with the highest diagnostic performance. The limitations in standardizing routine FEC tests are highlighted, and the importance of equalization of FEC results is emphasized for promoting uniformity in the implementation of parasite control guidelines.

## 1. Introduction

Evidence-based targeted anthelmintic treatment programs to control equine helminthic infections are gaining momentum, largely in response to reports of widespread anthelmintic resistance in cyathostomins (small strongyles) [1,2]. Once considered inconsequential to equine health, cyathostomins are now recognized for their pathogenic potential and their ability to develop resistance to many commonly used anthelmintics [3,4]. Small strongyles infect all age groups of horses, although infection intensity is greater in young and in some percentage of adult horses [5]. Anthelmintic interventions to disrupt the strongyle life cycle remain the mainstay for control of equine intestinal strongylosis. Designed to prevent infection with the dreaded large strongyle, *Strongylus vulgaris*, more than five decades of suppressive control programs have triggered the unexpected emergence of anthelmintic resistance in cyathostomins [2,6]. Despite the continued success of past programs in keeping *S. vulgaris* out of managed equine herds, the current need to mitigate resistance in small strongyles has led to increased awareness of evidence-based control programs [7]. The American Association of Equine Practitioners (AAEP) has framed the parasite control guidelines that explain the current scenario of anthelmintic resistance in equine herds in the USA and the plans to mitigate it [8]. The practices of blanket treatment or interval treatment regimens based on egg reappearance in the feces of all animals in a herd have been recognized as the cause of the emergence of drug resistance [7]. Cyathostomins are documented in the USA to be resistant to major classes of drugs on the market. Resistance is widespread to benzimidazoles (fenbendazole, oxibendazole), common in pyrimidines (pyrantel), and indicated for macrolide lactones (ivermectin, moxidectin) [9]. Benzimidazoles are now considered a failed group of drugs to treat cyathostomosis [2]. For macrolide lactones, the egg reappearance period (ERP), an indicator of efficacy, has been substantially decreased. For example, ERP for moxidectin was down from 16 to 22 weeks when the drug was introduced to the current low of 10–12 weeks [8].

Evidence-based targeted treatment programs are designed following the well-known concept of an over-dispersed parasite population. This means that only a small proportion of horses in a herd (15–30%) harbor large parasite burdens and are responsible for most (80%) of the eggs shed on pasture (8, 10). Horses that shed 0–200 eggs per gram of feces (EPG) are categorized as low shedders (50–75% of the herd), 201–500 EPG as moderate shedders (10–20%), and >500 as high shedders (15–30%) [8]. Targeted treatment of heavy shedders would promote parasite refugia to mitigate resistance, prevent unwarranted anthelmintic exposure in low shedders, and reduce the cost of the anthelmintic program [10,11,12]. The success of any evidence-based program depends on the availability of accurate fecal egg count (FEC) tests [13]. Fecal egg counts are required to designate horses based on their egg-shedding potential and to monitor anthelmintic efficacy by the fecal egg count reduction test (FECRT). Intestinal strongyle egg count is essentially done in two ways: (1) dilution egg count, and (2) concentration egg count [14]. The former ‘estimates’ eggs present per gram of feces while the latter concentrates eggs present in one gram of feces and earnestly ‘enumerates’ them. Modified McMaster and Mini-FLOTAC techniques are examples of the former, and the Wisconsin floatation test for the latter [8,13,15,16]. No two tests that work on either of the above two principles have similar diagnostic performance [17]. In fact, no consensus has been reached to designate a FEC test as a gold standard to quantify equine intestinal strongyle eggs that generally range from 0 to 2500 EPG in feces [10]. Numerous modifications and variations exist for FEC tests to accommodate practical and pragmatic considerations of the end-user. Several factors may impact the accuracy of FEC analysis in horses: individual differences in egg shedding, over-dispersion in feces, sampling, and storage practices. However, the most important is the type of FEC method used [13]. Dilution techniques tend to overestimate intestinal strongyle egg counts [18]. Regression analysis is generally done to analyze the usefulness of a test for egg counts [19,20,21]. In the past, purified strongyle eggs have been used for such studies, although on a smaller scale [18]. However, it is not feasible to procure the necessary numbers of purified eggs for large-scale studies. In this regard, polystyrene beads with a specific gravity (SPG) of 1.06 similar to the SPG of strongyle eggs (average 1.055; range 1.03–1.10) can be used as a proxy [14,22]. Deming regression analysis is used for method comparison [23].

This study aimed to compare the performance of various FEC tests, by both the bead-spiked and equine intestinal strongyle positive fecal samples, to determine their usefulness for evidence-based anthelmintic treatment programs in equine production.

## 2. Materials and Methods

### 2.1. Fecal Samples

Equine fecal samples were mainly collected from forty Thoroughbred horses of the research herd maintained at the Baker Institute of Cornell University. The herd was maintained and remained as experimental animals for protocols approved by the Institutional Animal Care and Use Committee (IACUC) of Cornell University. In a prior study performed over 12 months during 2017–2018, all animals in the herd were categorized as either low (0–200 EPG), moderate (201–500 EPG), or heavy (>500 EPG) shedders, as per the AAEP guidelines. Fecal samples from a single low shedder were used for repeatability analysis. Five other low shedders were used for regression analysis while other horses were used for the experiments performed for Section 3. Equine fecal samples submitted to the Cornell Animal Health Diagnostic Center (AHDC) or procured through Cornell Ambulatory Clinics were also used.

### 2.2. Polystyrene Beads

Polystyrene microspheres (red-colored beads, 1.06 SPG, and 45 µm diameter) were procured from Phosphorex, Inc., Fall River, MA, USA, as 1.0 g dry powder. A stock solution was prepared by dispensing a scoop of polystyrene beads using a lab spatula (0.1 in × 0.2 in) in 1 mL of distilled water and further diluted in 1.5 mL of 10× PBS containing five drops of 0.1% Tween 20 and sodium azide. A working stock was prepared by titrating and counting beads under the compound microscope so that every 50 µL of the working solution contained an estimated 2080 ± 134 beads. Initial assessments confirmed that the beads float in 1.20 SPG NaCl, 1.33 SPG NaNO_3_, 1.33 SPG sugar, and 1.18 SPG ZnSO_4_ floatation media.

#### 2.2.1. Validation of Bead Method: Recovery of Beads from the Fecal Matrix

Compatibility for a fecal matrix was tested by spiking polystyrene beads in fecal sediment from 6 different horses (with a known EPG of zero), and their recovery through floatation solutions was analyzed (Figure 1). For this pilot study, 12.5 μL (520 ± 33 beads) of the working stock solution was spiked to sediment obtained after straining a gram of horse feces in a tea strainer. Spiked sediments from each horse placed in two different centrifuge tubes were mixed separately with ZnSO_4_ (1.18 SPG) and sugar (1.33 SPG). The beads were retrieved under a coverslip, using the protocol as that of the modified Wisconsin double centrifugation floatation technique (Refer Section 2.3.3).

#### 2.2.2. Validation of Bead Method: Standardized Bead Dilutions

To overcome the difficulty of manual titrations of bead stock solution to obtain the desired concentrations, the polystyrene beads were sorted using BioSorter (large object flow cytometer) at the core facility of the Cornell Institute of Biotechnology. Briefly, 0.1 g of beads was dispensed in 30 mL of deionized distilled water containing Tween20 and blended well to avoid clumps. Using the built-in program of the BioSorter, the required numbers of beads (63, 125, 250, and 500) were collected in vials. As individual beads were sorted as droplets, the desired numbers of beads were collected as follows: 63 beads in 0.1 mL, 125 beads in 0.5 mL, 250 beads in 1.0 mL, and 500 beads in 2.0 mL. Sorted beads remained at 4 °C until further use. The bead number in three aliquots for each concentration was verified by dispensing the precipitated beads onto a glass slide and counting using a light microscope. The median for bead count was determined.

#### 2.2.3. Validation of Bead Method: Repeatability Assay

This study was designed to analyze the consistency of bead standard (125, 500, and 1000 beads) recovery among replicates and to prove the usefulness of these beads as a proxy for FEC using various quantitation protocols. Fecal aliquots from a single horse, designated as a low shedder (EPG = 0 by modified Wisconsin method), were spiked with the bead standards, and their recovery was analyzed by fecal quantitation tests (n = 12, Table 1) with each test replicated 12 times. Thorough cleaning of the apparatus between reuses was ensured to avoid bead contamination. A total of 432 runs (1 animal × 12 fecal replicates × 3 bead standards × 12 methods) were performed to obtain data to analyze statistical significance for repeatability assay. The coefficient of variation (CV%) for three bead standards for each test was assessed.

### 2.3. Fecal Egg Count Methodologies

Three techniques (modified Wisconsin, modified McMaster, and Mini-FLOTAC) and their four variants based on the choice of floatation solution used (ZnSO_4_ 1.18 SPG, sugar 1.33 SPG, NaCl 1.20 SPG, NaNO_3_ 1.33 SPG) were included in this study. The combination of three techniques and four variants resulted in a total of 12 different tests to assess their diagnostic performance for counting beads and intestinal strongyle eggs in fecal samples. The technical process is the same for estimating either beads or intestinal strongyle eggs, except for spiking the desired number of beads to fecal samples as outlined below. All techniques were performed by trained technicians at the AHDC parasitology lab.

#### 2.3.1. Modified McMaster Technique

The desired number of beads was added to 2 g of fecal samples in a wax paper cup. The vials were rinsed minimally with tap water using a jetwash bottle with a fine stream nozzle to dispense all beads into the wax cup. The completeness of dispensing all beads from a vial to the fecal sample was assessed stereo-microscopically, and no remaining beads were observed. Using a serological pipette, 28 mL of desired float solution was dispensed in the wax cup, and fecal samples with beads were mixed well using a tongue depressor. The content was sieved through a metal strainer and pressed with the tongue depressor to extract as much filtrate as possible. The filtrate was mixed thoroughly and dispensed to fill both chambers of the McMaster counting slide using a disposable transfer pipette. After 5 min of wait time, the beads or eggs in the chambers were counted under a compound microscope, and the estimate in a gram of fecal sample was determined based on an established formula {[Number of beads or eggs × (30 mL/0.3 mL)]/2 g} [24].

#### 2.3.2. Mini-FLOTAC Technique

This technique was based on a published protocol using the commercially available kit [24,25]. Briefly, 2 g of feces was placed in the Fill-FLOTAC cup, to which the desired number of beads were dispensed. Vials were cross-checked stereo-microscopically to confirm the dispensing of all beads. The sample was homogenized in 38 mL of the desired float solution and loaded into both cassettes of Mini-FLOTAC. After 10 min, the key on the device was rotated to a 90° angle and the beads/eggs were counted under a compound microscope. The count from both cassettes was multiplied by a factor of 10 to derive the estimate of beads/eggs per gram of feces.

#### 2.3.3. Modified Wisconsin Double Centrifugation Floatation Technique

To one gram of fecal sample in a wax paper cup, the desired number of beads from a vial was dispensed and mixed in 15 mL of tap water. Vials were cross-checked to rule out incomplete dispensing. After sieving through a metal strainer, the filtrate was dispensed into a 15 mL glass tube and centrifuged at 2000 rpm for 10 min. The supernatant was poured off and the sediment was homogenized with ≈15 mL of the desired float solution using a wooden applicator stick. The floatation solution was added to the brim of the glass tube to form a slightly positive meniscus before a glass coverslip (22 × 22 mm) was placed onto it. After centrifuging at 2000 rpm for 10 min, the coverslips were removed and placed on a glass slide. Beads or eggs under the coverslips were counted manually using a compound microscope (100× magnification) and expressed as beads/eggs per gram of feces [14,24].

### 2.4. Regression Analysis

Deming regression was performed to identify a test’s suitability to quantify the intestinal strongyle eggs present in a given fecal sample over a plausible range (low to high). Deming regression is a widely accepted method for this purpose [23] due to the suitability of this method when there is measurement error in both variables of the regression. Fecal aliquots of five different horses designated as low shedders (EPG = 0 as determined by modified Wisconsin) were spiked separately with 63, 125, 250, 500, and 1000 beads, and their recovery was analyzed and plotted against the predicted bead count average of 2 replicates. The choice of the bead standards (63 to 1000) reflects the FEC ranges indicated in AAEP guidelines for designating horses based on their egg-shedding potential (0–200 EPG as low; 201–500 EPG as moderate; and >500 EPG as high shedders). The coefficient of determination (R^2^) of the linear fit indicates the suitability of that particular test to perform linearly over the biologically important range of intestinal strongyle egg quantitation. Subsequently, a correction factor (CF) was deduced from the Deming regression forced through a 0 intercept for tests that showed a sufficiently high coefficient of determination in the Deming regression (R^2^ > 0.95).

### 2.5. Method Agreement

Firstly, fecal samples from 40 different horses were analyzed to verify whether FEC differed between the quantitation techniques with R^2^ > 0.95. Secondly, the CF was applied. Pearson correlation between each pair of tests was determined. FEC results were then categorized into low, moderate, and high shedders. Contingency analysis was performed to determine the kappa agreement before and after adjusting with CF.

## 3. Results

### 3.1. Bead Recovery from the Fecal Matrix

Polystyrene beads (520 ± 33 in 12.5 µL) spiked into fecal sediments of six different horses were retrieved under coverslips after mixing and floatation in ZnSO_4_ (1.18 SPG) and sugar (1.33 SPG). Spiked beads were detected using light microscopy in 100× magnification (Figure 1). ZnSO_4_ solution retrieved an average of 452 beads (range 250–598) from six horses, whereas sugar solution retrieved 552 beads (range 528–618).

### 3.2. Repeatability of Bead Standard Recovery

Results of the repeatability assay on 12 replicates for each test for selected bead standards (125, 500, and 1000) are depicted in Table 1, showing the average number of beads recovered (95% CI), percentage recovery (95% CI), and their coefficient of variation.

The average CV% for the modified Wisconsin test variants NaCl 1.20 SPG, NaNO_3_ 1.33 SPG, sugar 1.33 SPG, and ZnSO_4_ 1.18 SPG were 40.25, 44.2, 28.14, and 32.49, respectively. The average CV% for the modified McMaster test variants (same order as above) were 45.5, 40.48, 41.59, and 29.19. Similarly, for Mini-FLOTAC, the average CV% were 21.45, 23.95, 28.31, and 26.74. Among the 12 tests, Mini-FLOTAC NaCl 1.20 SPG had the lowest CV%, indicating its precision in the mean number of beads recovered for the three bead standards. In general, the Mini-FLOTAC-based variants had the lowest CV%, whereas the McMaster variants had the highest.

The average percentage recovery of three bead standards for the Mini-FLOTAC method variants was 76.3% for NaCl 1.20 SPG, 66.4% for NaNO_3_ 1.33 SPG, 71.26% for sugar 1.33 SPG, and 65% for ZnSO_4_ 1.18 SPG. The average percentage recoveries of beads for the modified McMaster method variants were 95.7% for NaCl 1.20 SPG, 51.1% for NaNO_3_ 1.33 SPG, 90.3% for sugar 1.33 SPG, and 92.06% for ZnSO_4_ 1.18 SPG. Some variants of the modified McMaster method overestimated the spiked bead standards. For example, a replicate of 125 spiked beads was estimated as 350 by NaCl 1.20 SPG. Similarly, a replicate of 1000 beads was estimated as 1100 by both NaCl 1.20 SPG and sugar 1.33 SPG. A significant impact of floatation solution in bead recovery (51.1%) was noted for the NaNO_3_ 1.33 SPG variant compared to the other three variants of modified McMaster. The average percentage recoveries of beads for the modified Wisconsin method variants were 44.63% for NaCl 1.20 SPG, 47.63% for NaNO_3_ 1.33 SPG, 62.8% for sugar 1.33 SPG, and 46.63% for ZnSO_4_ 1.18 SPG.

### 3.3. Deming Regression Analysis

Bivariate analysis and linearity of the regression curve were determined by Deming regression (Figure 2). The regression curve that passes through ‘zero’ may indicate a perfect slope. None of the tests generated a regression curve that passed through zero, which was desirable to identify a single correction factor. Therefore, the regression equation was calculated by forcing a 0 intercept into the model. None of the methods and their variants had an R^2^ = 1, or a slope = 1. All variants of Mini-FLOTAC and modified Wisconsin NaNO_3_ 1.33 SPG bead counts followed a linear fit with R^2^ > 0.95. The bead standard replicates for the modified McMaster variants dispersed from the regression curve, causing a lower R^2^. For the modified Wisconsin variants, bead standard replicate values were tighter for the lower number of spiked beads (63, 125, and 250) but dispersed from the regression curve for higher numbers (500 and 1000). Similarly, Mini-FLOTAC variants (NaCl 1.20 SPG and sugar 1.33 SPG) as well as the modified McMaster variants (sugar 1.33 SPG and ZnSO_4_ 1.18 SPG) had slope < 1.3 (closer to 1).

R^2^ and slope of linear regression were analyzed to determine the goodness-of-fit for each of the 12 tests (Table 2). A value closest to 1 for both parameters (R^2^ and slope) indicates good fit. Four tests had a slope < 1.30 (modified McMaster sugar 1.33; Mini-FLOTAC sugar 1.33; modified McMaster ZnSO_4_; and Mini-FLOTAC NaCl 1.20). Five tests had R^2^ > 0.95 (all variants of Mini-FLOTAC and modified Wisconsin NaNO_3_).

### 3.4. Correlation of Best Methods in Fecal Samples

The Pearson correlation for five method modifications with R^2^ > 0.95 was tested using 40 fecal samples (Table 3). The correlations were generally very high, and the ones above 0.90 are highlighted in Table 3. The Mini-FLOTAC NaCl 1.20 SPG as well as ZnSO_4_ 1.18 SPG had the highest number of correlations ≥ 0.90 with other methods and variants. Mini-FLOTAC ZnSO_4_ had the highest correlation (0.9682) with Mini-FLOTAC NaCl.

### 3.5. Agreement between Methods for Classification of Shedding Category before and after Adjustment

The raw estimates as well as the corrected estimates of FEC from 40 horses were categorized based on strongyle egg shedding categorization (based on the cut-offs as per AAEP guidelines (see Section 2.1)). Contingency analysis for FEC categories, low (L), moderate (M), and heavy (H), was performed with raw and adjusted FEC estimates (Figure 3). The median (range) kappa value across all comparisons was 0.54 (0.16–0.88). The adjustments equalized the categories to some extent though not perfectly to a median (range) kappa of 0.67 (0.51–0.85).

## 4. Discussion

Polystyrene beads were used as a proxy for intestinal strongyle eggs to overcome the difficulty of procuring extracted/purified eggs for such a large-scale study. Initial validation to assess beads’ compatibility with horse fecal matrix and their ability to float in various floatation solutions of differing SPG proved their usefulness for such a study. One major hurdle was aliquoting the desired number of beads via manual titrations as performed in prior studies using strongyle eggs [26,27]. However, bead sorting through the large object flow cytometer (BioSorter) obtained aliquots of appropriate concentrations needed. The efficiency of BioSorter was verified manually by counting the sorted beads from a subset of aliquots that yielded satisfactory results. No residual beads observed stereo-microscopically in the vials after transferring them into fecal samples proved the completeness of the dispensing process.

An initial assay to determine the repeatability of all three FEC methods and their variants was performed using three select bead standards (125, 500, and 1000). Accurate recovery of spiked beads and minimal coefficient of variations (CV%) among replicates must indicate the repeatability of the assays. In general, low repeatability was observed for all 12 tests as none had CV% closer to zero or recovered 100% of the spiked beads. Mini-FLOTAC had a comparatively better CV% (average < 28.3) and bead recovery average of >65% among the three methods. Our result corroborated the findings of other studies that confirmed less variance for Mini-FLOTAC [28,29,30]. The modified McMaster had a CV% of >40 for all variants except ZnSO_4_ and improved recovery of >90% for all variants except NaNO_3_. The superiority of the modified McMaster technique in the apparent recovery of a higher number of beads is attributed to its bigger multiplication factor [29]. All variants of modified Wisconsin except sugar 1.33 had CV% > 32 and average recovery as low as 44.6%. This differed from a previous study that showed that the Cornell–Wisconsin method with NaCl 1.20 SPG had a low CV% [30]. A larger CV% associated with the modified Wisconsin and modified McMaster methods may negate the suitability of these techniques for a reliable bead count. The processing steps inherent to these two methods and the choice of floatation solution might have influenced the low and inconsistent recovery of beads. For example, NaNO_3_ 1.33 SPG negatively affected bead recovery using the modified McMaster method but sugar 1.33 SPG positively impacted the modified Wisconsin method. Nevertheless, the Mini-FLOTAC method seems less influenced by the choice of floatation solution and has better repeatability parameters, indicating its reliability for bead counts. The processing of fecal samples using Fill-FLOTAC associated with Mini-FLOTAC is credited for its better diagnostic performance [30,31].

The regression curve assessment (linear behavior) of the 12 tests to recover the bead standards proved that no test was linear as the slopes were not passing through zero (Figure 2). The four variants of Mini-FLOTAC had superior performance in bead recovery, especially the sugar 1.33 variant, in which most replicates congregated closer toward the slope and performed best in the Deming regression based on the highest R^2^ (0.977) and slope (1.2057563) that are closer to 1. However, most replicates of Mini-FLOTAC underestimated the bead count. For the modified McMaster method, most replicates tended to disperse away from the slope. However, many replicates overestimated spiked beads, indicating a potential threat of false counts [29]. The modified Wisconsin method variants performed better in counting the lower bead standards whereas the higher ones tended to disperse away from the slope. All variants of the modified Wisconsin method underestimated the bead counts, similar to the study performed with spiked strongyle eggs [30].

Regression analysis (coefficient of determination) determined the suitability of each of the 12 tests to recover beads mimicking the plausible range of strongyle FEC referred to in the AAEP parasite control guidelines. All variants of Mini-FLOTAC and the NaNO_3_ 1.33 variant of modified Wisconsin had R^2^ > 0.95. Other tests had poor R^2^ values. Mini-FLOTAC sugar 1.33 performed better as the R^2^ value, and the slope was closer to but not equal to 1. This may preclude it from being designated as the gold standard test. Despite its good repeatability, Mini-FLOTAC variants were unable to estimate 100% of the spiked beads. Nonetheless, the correction factor (CF) determined for each test can be applied to the raw estimate to yield an adjusted estimate that should be closer to the expected count.

An accurate FEC test (i.e., estimates correctly) does not need a CF. A FEC test that is not sensitive (i.e., underestimates consistently) but precise (i.e., estimates sharply) needs CF. Indeed, CF was intended for tests with high R^2^ (>0.95 < 1) to help standardize FEC. We applied CF to FEC from 40 horse fecal samples, determined by five different method modifications with R^2^ > 0.95. Pearson correlation on the adjusted FEC pointed out the Mini-FLOTAC method variants (ZnSO_4_ and NaCl) having the most correlations with others. Although the five method variants correlated positively, the CF adjustments did not equalize the EPG counts (as shown by the only improved kappa value). Alternatively, the agreement between the five methods based on egg-shedding categories (low, moderate, heavy) was determined as per AAEP guidelines, before and after CF adjustments. Interestingly, the FEC categories equalized to some extent between the five tests after CF adjustments. Equalizing the FEC based on AAEP egg-shedding categories allows for a standardization of FEC methods. Further, this will facilitate meaningful approaches for targeted control of intestinal strongyle infection and effective management of anthelmintic resistance.

## 5. Conclusions

The diagnostic performance of FEC methods (modified Wisconsin, modified McMaster, and Mini-FLOTAC) and their variants (based on the choice of floatation solutions: NaCl 1.20 SPG, NaNO_3_ 1.33 SPG, sugar 1.33 SPG, and ZnSO_4_ 1.18 SPG) differed considerably. Using polystyrene beads as a proxy for strongyle eggs proved that no method modifications behaved linearly in detecting bead standards mimicking the plausible range of FEC for horses. Regression analysis pointed out five tests (Mini-FLOTAC NaCl 1.20 SPG, Mini-FLOTAC NaNO_3_ 1.33 SPG, Mini-FLOTAC sugar 1.33 SPG, Mini-FLOTAC ZnSO_4_ 1.18 SPG, and modified Wisconsin NaNO_3_ 1.33 SPG) with R^2^ and slope closer to 1 as suitable for FEC. Despite their suitability, the raw FEC estimate differed, and a correction factor determined for each test based on regression analysis was applied to standardize methodologies. Furthermore, equalizing the FEC results based on egg-shedding categories (low, moderate, and heavy) outlined in AAEP guidelines helped with standardization efforts. Selecting a suitable test with the application of the CF is the key to determining FEC. Adjusted FEC should form the basis for controlling and management of equine strongylosis.

## Figures and Tables

**Figure 1 pathogens-12-01283-f001:**
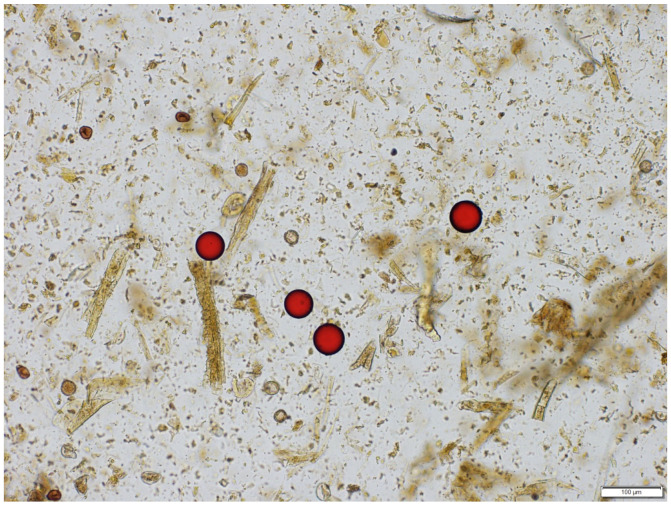
Photomicrograph of recovery of red polystyrene beads spiked in fecal sediment from 6 different horses with a known EPG of 0, obtained after removal of coarse fecal material and floated subsequently in sugar 1.33 SPG. Scale bar = 100 µm.

**Figure 2 pathogens-12-01283-f002:**
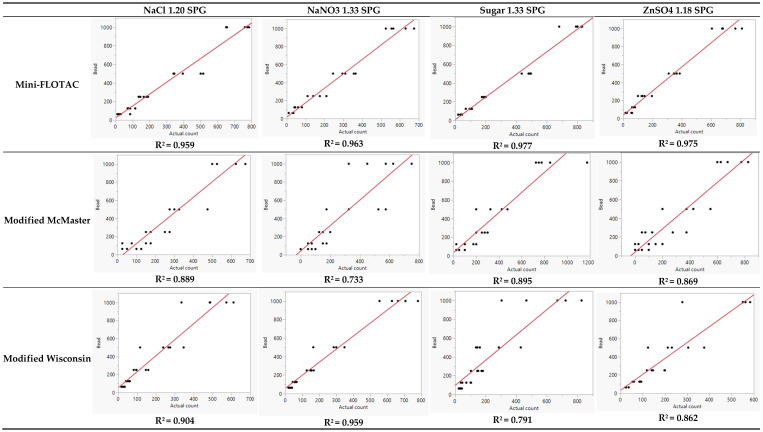
Linear behavior of bead standard recovery by various fecal quantitation tests. Bead standards (63, 125, 250, 500, and 1000) in the y-axis plotted against the actual count estimate determined for each of the 12 tests on 5 replicates in the x-axis.

**Figure 3 pathogens-12-01283-f003:**
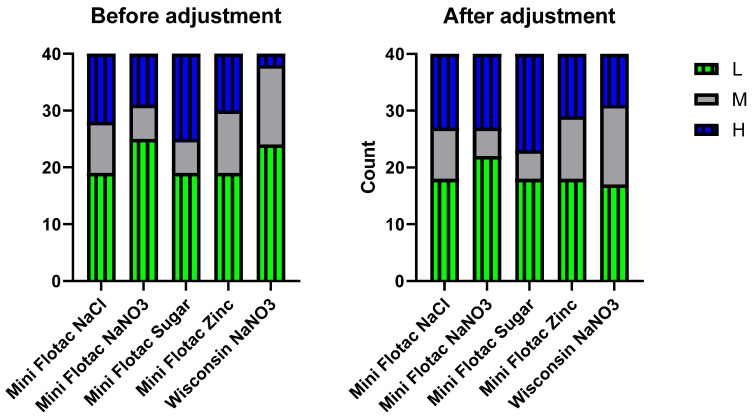
Agreement of FEC categorization results (low 0–200; medium 201–500; heavy > 500) between the 5 selected methods (x-axis) before and after application of CF as determined by kappa analysis. Forty fecal samples are indicated in the y-axis.

**Table 1 pathogens-12-01283-t001:** Assessment of repeatability and recovery of bead standards (125, 500, and 1000) by three fecal quantitation methods and four modifications/variants based on the preference of commonly used floatation solution.

		Method											
Modification	Bead Input	Modified Wisconsin				Modified McMaster				Mini-FLOTAC			
		Average^1^ (95% CI)	Min–Max	CV%	% Recovery (95% CI)	Average (95% CI)	Min–Max	CV%	% Recovery (95% CI)	Average (95% CI)	Min–Max	CV%	% Recovery (95% CI)
**NaCl 1.20**													
	125	59^a^ (20–98)	26–99	34.0	48 (17–78)	163^ab^ (124–201)	50–350	65.8	130 (99–161)	98^b^ (60–137)	50–170	34.1	79 (48–11)
	500	248^a^ (190–306)	145–371	36.5	50 (38–61)	429^b^ (371–487)	200–700	30.4	86 (74–97)	403^b^ (346–461)	290–540	15.5	81 (69–92)
	1000	367^a^ (246–488)	79–693	50.3	37 (25–49)	713^b^ (592–834)	200–1100	40.4	71 (59–83)	695^b^ (574–816)	560–960	14.8	70 (57–82)
	**Total^2^**			**40.3**	**44 (39–51)**			**45.5**	**96 (77–115)**			**21.5**	**76 (70–82)**
**NaNO_3_ 1.33**													
	125	60 (44–76)	25–96	40.9	48 (35–61)	58 (42–74)	0–150	61.5	47 (34–60)	64 (48–80)	30–90	29.3	51 (39–64)
	500	247^ab^ (191–303	84–407	45.3	50 (38–61)	246^b^ (190–302)	100–400	40.2	49 (38–60)	343^b^ (287–399)	220–500	20.8	69 (57–80)
	1000	452^a^ (352–552)	131–774	46.6	45 (35–55)	575^a^ (475–675)	400–700	19.8	58 (48–68)	792^b^ (692–892)	580–1050	21.8	79 (69–89)
	**Total^2^**			**44.2**	**48 (41–54)**			**40.5**	**51 (44–58)**			**24.0**	**66 (60–73)**
**Sugar 1.33**													
	125	68 (40–96)	25–103	44.3	54 (32–76)	125 (97–153)	50–250	55.3	100 (78–122)	75 (47–103)	30–130	42.7	60 (38–82)
	500	367 (294–440)	285–421	13.0	73.3 (59–88)	458 (385–532)	250–950	42.3	92 (77–106)	392 (319–465)	230–510	20.9	78 (64–93)
	1000	609 (501–716)	140–769	27.2	61 (50–71)	792 (685–899)	450–1100	27.2	79 (65–86)	755 (648–862)	420–1060	21.3	76 (65–86)
	**Total^2^**			**28.1**	**63 (57–69)**			**41.6**	**90 (77–104)**			**28.3**	**71 (64–78)**
**ZnSO_4_**													
	125	67 (40–93)	46–85	18.4	53 (32–75)	129 (103–156)	50–250	55.9	103 (82–125)	79 (53–106)	30–120	35.5	63 (42–85)
	500	217^a^ (175–258)	106–356	40.7	43.3 (35–52)	479^b^ (438–521)	400–550	10.4	96 (88–104)	285^a^ (244–326)	160–370	23.8	57 (49–65)
	1000	434^b^ (339–530)	286–834	38.4	43.4 (34–53)	771^a^ (676–866)	450–1000	21.3	77 (68–87)	747^a^ (651–842)	490–1070	20.9	75 (65–84)
	**Total^2^**			**32.5**	**47 (42–52)**			**29.2**	**92 (80–104)**			**26.7**	**65 (59–71)**

**Table 2 pathogens-12-01283-t002:** Determining the R^2^ and slope of regression for various fecal quantitation tests using bead standards spiked in fecal samples. Highlighted in red are slopes. The formula for correcting bead underestimation was derived based on regression coefficients (purple) when forcing the slope intercept through 0 and multiplied (*) with raw bead estimate.

		NaCl 1.20 SPG	NaNO3 1.33 SPG	Sugar 1.33 SPG	ZnSO4 1.18 SPG
**Mini-FLOTAC**	R^2^	0.959	0.963	0.977	0.975
	Adjusted = Intercept + Slope * Raw estimate	Adjusted = 23.971124 + 1.277684 * Raw estimate	Adjusted = 17.217636 + 1.6006152 * Raw estimate	Adjusted = 8.7513714 + 1.2057563 * Raw estimate	Adjusted = 38.400997 + 1.3389532 * Raw estimate
	Adjusted = (Slope intercept forced through 0) * Raw estimate	Adjusted = 1.323388 * Raw estimate	Adjusted = 1.641578 * Raw estimate	Adjusted = 1.2213087 * Raw estimate	Adjusted = 1.4152846 * Raw estimate
**Modified McMaster**	R^2^	0.889	0.733	0.895	0.869
	Adjusted = Intercept + Slope * Raw estimate	Adjusted = −48.56889 + 1.7037847 * Raw estimate	Adjusted = 41.524645 + 1.4011148 * Raw estimate	Adjusted = 35.072876 + 1.0747778 * Raw estimate	Adjusted = 39.778453 + 1.2377991 * Raw estimate
	Adjusted = (Slope intercept forced through 0) * Raw estimate	Adjusted = 1.5808012 * Raw estimate	Adjusted = 1.4993345 * Raw estimate	Adjusted = 1.132996 * Raw estimate	Adjusted = 1.3149801 * Raw estimate
**Modified Wisconsin**	R^2^	0.904	0.959	0.791	0.862
	Adjusted = Intercept + Slope * Raw estimate	Adjusted = 52.847268 + 1.7800315 * Raw estimate	Adjusted = 58.813301 + 1.4057923 * Raw estimate	Adjusted = 94.50772 + 1.3474268 * Raw estimate	Adjusted = 30.745896 + 1.7494563 * Raw estimate
	Adjusted = (Slope intercept forced through 0) * Raw estimate	Adjusted = 1.9250685 * Raw estimate	Adjusted = 1.5296038 * Raw estimate	Adjusted = 1.5573829 * Raw estimate	Adjusted = 1.8338515 * Raw estimate

**Table 3 pathogens-12-01283-t003:** Pearson correlation coefficient for FEC methods and their variants determined using fecal samples from 40 different horses. Highlighted in green are those with correlation coefficients ≥ 0.95, and those in yellow ≥ 0.90 ≤ 0.95.

	Mini-FLOTAC NaCl	Mini-FLOTAC NaNO_3_	Mini-FLOTAC Sugar	Mini-FLOTAC ZnSO_4_	Wisconsin NaNO_3_
Mini-FLOTAC NaCl	1	0.932	0.913	0.968	0.778
Mini-FLOTAC NaNO_3_		1	0.886	0.943	0.701
Mini-FLOTAC Sugar			1	0.914	0.751
Mini-FLOTAC ZnSO_4_				1	0.781
Wisconsin NaNO_3_					1

## Data Availability

Not applicable.

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
