# Peer review of "Evaluation of Fecal Egg Count Tests for Effective Control of Equine Intestinal Strongyles"

_pathogens, 2023, doi:10.3390/pathogens12111283_

Round 1

Reviewer 1 Report

Comments and Suggestions for Authors

General comments and suggestions:

There are various fecal egg count (FEC) techniques with many modifications and variations in use, but none is identified as a gold standard and methods comparison studies are lacking. This study identified methodologies with the highest diagnostic performance and the limitations to standardizing routine FEC tests to promote uniformity in implementing American Association of Equine Practitioners (AAEP) parasite control guidelines. The manuscript is scientifically sound and the findings are relatively explained. While the writing should improve and the specific comments and suggestions are as follows.

Specific comments:

1.     Keywords. It is better to delete the words which are not essential.

2.     “2.5 Fecal samples” can be changed to the first paragraph as “2.1”.

3.     It is suggested to show the content in Table 1 in words but not a table.

4.     The scale bar in Figure 1 is not clearly enough and should be improved.

5.     Discussion. It is unscientific not to cite references here. And this section should not show results in too many words. Please revise.

6.     There are some format or spelling mistakes which should be check and correct.

(1)   Tables should be presented as three-line.

(2)   Please check and correct the writing of the MS such as “(R2>0.95” in 2.3, and “um” should be “μm”.

(3)   “P” of “P value” should be italic.

(4)   Please unify the format of the Reference according to the requirement of the journal. For example, whether the journal name needs to be in abbreviation and the case of content words.

.

Author Response

File attached. 

Reviewer 2 Report

Comments and Suggestions for Authors

Very well written manuscript comparing FEC methods for horses, an interesting and relevant topic. In general, this manuscript could benefit from the addition of more recent references (i.e., within last 5 years) from research labs in the US that study this topic. Please consider adding such references in introduction and discussion.

Abstract:

It is mentioned that comparison studies are lacking. However, there are at least two handfuls in the literature. I would suggest modifying the sentence in the text. 

Given the amount of results in the body of the manuscript, please shorten the introduction of the abstract and add more significant results. A few examples of information from the results section that should be considered for the abstract: (1) "In general, Mini-FLOTAC based variants had the lowest CV% whereas McMaster variants had the highest." (2) "all variants of Mini-FLOTAC and modified Wisconsin NaNO3 1.33 SPG bead counts followed a linear fit with R2 >0.95. In contrast, the bead standard replicates for modified McMaster variants dispersed from the regression curve, causing a lower R2." (3) "Mini-FLOTAC ZnSO4 had the highest correlation (0.9682) with Mini-FLOTAC NaCl."

Minimal summary or discussion points mentioned in the current abstract. Please consider adding stronger takeaway points. One such statement could include, "The Mini-FLOTAC method seems less influenced by the choice of flotation solution and had better repeatability parameters indicating its reliability for bead counts."

Introduction: 

An updated review of anthelmintic resistance in equine nematodes is available: Nielsen MK. Anthelmintic resistance in equine nematodes: Current status and emerging trends. Int J Parasitol Drugs Drug Resist. 2022 Dec;20:76-88. Please consider adding it into the reference list and introductory section.

Please add a reference for this statement: "Benzimidazoles are now considered a failed group of drugs to treat cyathostomosis."

Some equine parasitologists may argue that there is a gold standard FEC method. Given this is subjective, please consider modifying this sentence to mention that among equine parasitologists, it does not appear a consensus has been made. 

For the aim of the study, please mention that the performances of the tests were analyzed by both the bead-spiked samples and equine strongyle samples. How it is currently written, it does not adequately describe the extent of the studies performed in this manuscript.

Methods:

2.1 Please consider adding a reference for this section, if available.

2.1.1 Please indicate whether or not the feces were strained prior to placement in the centrifuge tube.

2.1.3 Since the repeatability assay used a sample from a horse designated as a low shedder, can you please add the exact FEC from that sample used. Any variability, if present, added by a horse with low FEC should be known by the reader as they interpret the repeatability results.

2.2 Some of this information has been repeated from the introduction section. Is it all necessary or can it be made more concise?

2.2 Please cite a reference for the techniques used (e.g., Zajac et al., 2021). Need to add details on those who performed these techniques. Were they trained laboratory technicians and microscopists? How much training did they have?

2.2 There is no reference listed as [24] in list at end of document.

2.2.1 There is no reference listed as [25] in list at end of document.

2.2.1 Any wait time between filling McMaster and counting the beads? Please add this detail to the manuscript or cite the reference followed. 

2.2.2 There is no reference listed as [26] in list at end of document.

2.3 Please provide the exact FEC of the 5 horses used.

2.3 Need to close the parentheses at the end of the paragraph.

Results:

3.1 Not sure there is a way to say with certainty that the fecal matrix had no impact on polystyrene beads. Maybe more appropriate to say minimal.

3.2 Is this results or would it be more appropriate in methods?

Table 2: As a footnote in this table, please write out all abbreviations so the reader so not have to search through the text to interpret the table.

Table 2: Even though the goal was to have the same bead input for every variation along a row, it seems, based on averages and max, that there were many times the bead number counted was well over the intended initial bead input. Given this, I'm not sure the results can be interpreted correctly or taken seriously in the statistics. Maybe I'm not understanding correctly, but this seems like a major flaw. Is it an issue with the BioSorter? Or were beads retained by the reusable slides between reps... because the averages and max never went above the intended input when using the Wisconsin? Most other references cite Wisconsin as having the highest sensitivity and specificity among FEC techniques. In your response back to reviewers, please explain your answers thoroughly and include references, as appropriate.

Since samples were spiked with a known number of beads, any reason why accuracy and precision were not calculated?

3.4 (6 lines down): Correct punctuation around the word "however"

Figure 2: Any reason why R2 values are not present on each line? That would be helpful for the reader. If these are added to the figure, table 3 can become a supplemental file. 

The text that first cites Table 3 mentions that R2 and slope of linear regression were used to estimate the actual number of spiked beads, but the table only shows information about R2 and slopes. Is this is typo in the text?

Table 3: Correct the missing space between "when" and "forcing" on the last line.

Table 3: Without a legend, the use of an asterisk in the table, especially a mathematical table, appears to be signifying a multiplication factor. Please clarify. 

Discussion:

Throughout: This appears to be written more as a summary. It is a great summary, but this section warrants a discussion of the results in comparison to previous studies. Even though this is a novel method to determine repeatability, agreement, etc., there are other published studies that compare between fecal diagnostic tests. Please add in references throughout and draw comparisons where applicable. 

First paragraph: What about making sure no beads remained on the reusable slides? What methods were used to ensure clean slides each time?

Third paragraph: This sentence does quite make sense: "Barring a few replicates, the rest other underestimated the bead count indicating the low sensitivity of Mini-FLOTAC." Please rewrite. 

Third paragraph: "seems sensitive" and "low sensitivity" are mentioned. Without calculating these values and adding to the manuscript, I would suggest not using these words in the manuscript. 

Third paragraph: Please elaborate on potential reasons why the overestimation occurred for modified McMaster.

Fourth paragraph: "low sensitivity and less accuracy" again mentioned. Without calculating these values and adding to the manuscript, I would suggest not using these words in the manuscript.

Fifth paragraph: Are there other publications that have discussed an equalization factor to FEC or other quantitative techniques? If so, please add to this section. It would make your suggestion of adding an equalization factor much stronger if there are other publications stating its addition is helpful.

There is currently no mention of the practicality of some of the test combinations. For instance, given some of these solutions (i.e., sucrose) are not routinely used in a McMaster slide or a miniFLOTAC apparatus, I would suggest adding observations or recommendations about the proper cleaning of these instruments. Without such cleaning, it is assumed the technique would suffer long-term. Even if it leads to greater repeatability, if a method is not practical, it is unlikely to be used. 

Conclusion:

The final sentence is of great importance. This should be added to the abstract as well, if space allows. 

Author Response

File attached

Reviewer 3 Report

Comments and Suggestions for Authors

This is an interesting paper that deserves to be published in Pathogens after some improvement and editing is done to it.

- The title should reflect much better the objective of the work done, which is the comparison of techniques. "Evaluation of equine fecal egg count tests for accurate monitoring of anthelmintic resistance" is a bit too narrow as the results could also be applied to situations beyond AR, such as monitoring the parasite situation of herds in an epidemiological context throughout the different seasons, separate the individuals as low-medium-high shedders in a herd, etc.

One remarkable issue is that the authors have built a discussion based solely on their current findings and have not consulted the available literature. This issue alone faults the otherwise good manuscript almost to the point of rejection. Although the manuscript has novelty components (the use of beads replacing strongyle eggs, etc.) that might not have been used before, the fact remains that the results from comparing three different techniques need to be contrasted with the literature - several papers have been published through the years - even if such papers might not deal with equine nematodes. After all, most strongyle eggs from different host species are very similar in size and spg.

Besides the above comments, the authors should pay attention to the several comments made directly on the file.

Comments on the Quality of English Language

Author Response

File attached

Round 2

Reviewer 2 Report

Comments and Suggestions for Authors

Thank you for responding to each point and making edits as necessary. Look forward to seeing this manuscript published. 

Author Response

We thank the Reviewer for agreeing to all our responses. 

Reviewer 3 Report

Comments and Suggestions for Authors

Many thanks to the authors for their efforts in making the revised version of the paper a much better manuscript. A few other comments that are attached to the PDF document need to be attended.

Comments on the Quality of English Language

Author Response

We thank the Reviewer for all the useful suggestions and comments that helped to improve the quality of this manuscript. We will provide the editors with the original version of Table 1 and Table 2 to overcome the issues pointed.